# Effects of walking impairment on mental health burden, health risk behavior and quality of life in patients with intermittent claudication: A cross-sectional path analysis

**Farhad Rezvani** [ID]*, **Mara Pelt, Martin Härter, Jörg Dirmaier**

Department of Medical Psychology, University Medical Center Hamburg-Eppendorf, Hamburg, Germany

* f.rezvani@uke.de

## Abstract

### Introduction

Intermittent claudication is the leading symptom of peripheral artery disease (leg pain when walking). The present study investigates the extent to which walking impairment is associated with health-related quality of life, mental health and health risk behavior.

### Methods

A theory-based, cross-sectional path model was empirically examined using pre-intervention baseline data from a multicenter, randomized-controlled trial of patients with intermittent claudication (*PAD-TeGeCoach*). Data were available from 1 696 patients who completed a battery of questionnaires between April 14, 2018 and March 12, 2019, including measures of walking impairment (Walking Impairment Questionnaire), health-related quality of life (SF-12), mental burden (GAD-7, PHQ-9), nicotine- and alcohol-related risk behavior (Fagerström-Test, AUDIT-C). Sociodemographic characteristics and comorbid conditions were included in the postulated model a priori to minimize confounding effects.

### Results

Walking impairment was associated with an increase in depressive (β = -.36, p < .001) and anxiety symptoms (β = -.24, p < .001). The prevalence of depressive and anxiety symptoms was 48.3% and 35.5%, respectively, with female patients and those of younger age being at greater risk. Depressive symptoms were predictive of an increased tobacco use (β = .21; p < .001). Walking impairment had adverse effects on physical quality of life, both directly (β = .60, p < .001) and indirectly mediated through depressive symptoms (β = -.16, p < .001); and indirectly on mental quality of life mediated through depressive (β = -.43, p < .001) and anxiety symptoms (β = -.35, p < .001).

**Data Availability Statement:** All relevant data are included in the S1 File (SAV).

**Funding:** This study received funding from the German Innovation Fund (01NVF17013) of the

Federal Joint Committee (G- BA), the highest decision- making body of the joint self-government of physicians, dentists, hospitals and health insurance funds in Germany. The use of grant funds is monitored by the German Aerospace Center (Deutsches Zentrum für Luft- und Raumfahrt; DLR). Neither the G- BA nor the DLR was involved in the actual conduct of this work (ie, study design, data collection and analysis, decision to publish, or preparation of the manuscript).

**Competing interests:** The authors have declared that no competing interests exist.

## Discussion

The findings underscore the need for a comprehensive treatment strategy in patients with intermittent claudication. Measures to improve walking impairment (e.g. exercise training) are key to enhance quality of life and should be the primary treatment. As a key mediator of mental quality of life, depressive and anxiety symptoms should be addressed by rigorously including mental health treatment. Risky health behaviors should be approached by promoting behavior change (e.g. smoking cessation) as a secondary prevention of peripheral artery disease.

## Introduction

Peripheral Artery Disease (PAD) affects up to 240 million people worldwide and ranks as the third leading cause of atherosclerotic morbidity after coronary artery disease and stroke [1], making it one of the leading causes of disability [2–4]. The hallmark symptom in symptomatic PAD patients is Intermittent Claudication (IC), which is characterized by muscle pain in the legs during walking and which subsides with short periods of rest [5]. Confronted with walking impairment and reduced mobility, these symptoms reflect the progressive narrowing of the peripheral arteries and the resulting reduction of blood supply [6, 7] that, if left untreated, can result in amputation [8] and death [9].

There is growing evidence that psychosocial factors are linked to functional outcomes and play a substantial role in the pathogenesis of PAD [10]. Depressive and anxiety symptoms are highly prevalent among PAD patients [11–13], which in turn have been shown to be associated with poor walking ability [14–18] and severe leg symptoms (i.e. pain at rest) [12], putting PAD patients at greater (long term) risk for mortality and other adverse PAD events [13–17, 19–22]. Prior studies have also demonstrated the negative impact of PAD and IC symptoms on health-related quality of life (HRQoL) [23–26], which again was found to have prognostic value in predicting long-term survival in PAD patients [27]. Experiencing depressive symptoms are associated with significantly lower HRQoL compared with their non-depressed counterparts, highlighting the impact of mental health symptoms on the PAD patient's subjective appraisal of their health status [28, 29].

While the association of mental health status with PAD is well established, an important next step is to understand the underlying mechanisms (and directionality) to allow targeted interventions. Although still subject of vigorous debate [10], several potential behavioral mediators have been proposed; one of them, tobacco smoking, which is known as a potent risk factor for developing PAD, was also found to be a important factor in the relationship between depression and subsequent PAD events [19]. Likewise, mental distress has been reported to be associated with increased amounts of alcohol consumption [30], which in turn has been identified as a risk factor for PAD [31–36]. Overall, the current evidence indicates risky health behaviors as a pathway through which mental health burden is causing poor PAD outcomes [10].

The present study addresses several important questions for the psychosocial management of symptomatic PAD, with the goal to determine whether and how walking impairment is associated with diminished HRQoL and mental burden in PAD patients that suffer from IC. With increased interest in health status and patient-based measures in cardiovascular research, identifying their respective determinants is increasingly important. Furthermore, this study

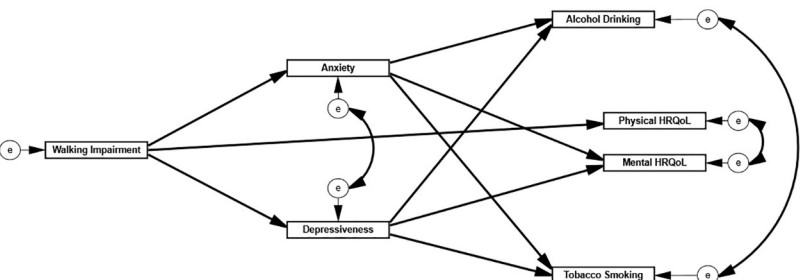

**Fig 1. Theory-based path analysis model regarding the influence of walking impairment on mental burden, health risk behavior, physical and mental HRQoL in PAD patients.**

also investigates the impact of mental health problems (i.e. depressive and anxiety symptoms) with risky health behaviors (i.e. alcohol and tobacco consumption), which in the long-term could be major drivers of a negative clinical PAD course [10]. A theory-driven, cross-sectional path model based on previous literature was therefore postulated (Fig 1) and empirically examined to explore the interrelationships among these constructs. The postulated model assumed an association between walking impairment and mental health burden (i.e. depressive and anxiety symptoms; Hypothesis 1), which in turn is linked to an increase in health risk behavior (i.e. tobacco smoking and alcohol drinking; Hypothesis 2). Moreover, walking impairment was hypothesized to have a direct, negative effect on physical HRQoL (Hypothesis 3) and an indirect, negative effect on mental HRQoL mediated by an increase of depressive and anxiety symptoms (Hypothesis 4). A better understanding of these relationships may foster the development of treatment strategies to improve HRQoL and emotional well-being in PAD patients, which may also indirectly have secondary benefits on PAD status.

## Methods

### Design

The present path analysis uses cross-sectional baseline data (i.e. pre-intervention) drawn from a larger data set that was collected as part of a multicenter, randomized-controlled trial (RCT) designed to test the effectiveness of a 12-month long telemedicine-guided home-based exercise program for patients with IC, *PAD-TeGeCoach* (CT.gov trial registration: NCT03496948). Methods of the *PAD-TeGeCoach* effectiveness trial are reported elsewhere in detail [37]. Ethical approval was granted by the ethics committee of the Medical Association of Hamburg. All patients provided written informed consent.

### Study population and recruitment

Participants were recruited using routinely collected health insurance data (electronic health records) from three German statutory health insurance funds (in German: *Gesetzliche Krankenversicherung*): Kaufmännische Krankenkasse, Techniker Krankenkasse, mhplus Krankenkasse. These three statutory health insurance funds together have approx. 12.1 million insured (TK: 10.5 million; KKH: 1.6 million; MH: 0.54 million) and cover 16.5% of all statutory insured citizens in Germany. Consequently, the patient population is likely to represent the PAD patient population presenting in a usual care setting.

Eligible patients were between 35 and 80 years old, and had a medically confirmed diagnosis of PAD at Fontaine stadium IIa (IC > 200 meters) or IIb (IC < 200 meters) within the last 36 months. Patients were excluded from the study if they had asymptomatic PAD within the

last 12 months (Fontaine stadium I) or rest pain within the last 36 months (Fontaine stadium III or IV). Patients with active or recent participation in other PAD intervention trials, medical conditions that contradict physical activity, cognitive disorders, severe mental disorders (including a clinical diagnosis of substance use disorder), suicidality, life-threatening illnesses, ongoing hospitalization, and heart failure (NYHA class III/IV) were also excluded.

The study population was derived from the *PAD-TeGeCoach* RCT; approximately 63 000 who met the inclusion criteria were identified as potential participants and were invited to participate. Of those, 1 982 elected to participate (recruitment rate 3.2%) and were randomized either into the intervention arm or the routine care group (see S1 Fig). There were 11 participants who withdrew after enrollment (data deletion request n = 1, randomized without informed consent n = 1, met exclusion criteria n = 8, lack of verification of PAD diagnosis n = 1).

## Measures

The data used for this study (i.e. baseline data from the *PAD-TeGeCoach* RCT) were collected between April 14, 2018 and March 12, 2019. Enrolled patients received a battery of self-administered questionnaires by mail (paper-pencil) and were asked to return them using a prepaid envelope. The participants could call the study team when they encountered problems completing the questionnaires. A total of 1 696 patients returned their study questionnaire, which falls within the usual range of mail surveys [38].

**IC symptoms: Walking Impairment Questionnaire (WIQ).** The Walking Impairment Questionnaire (WIQ) is a well-established instrument of assessing walking impairment for different degrees of difficulty across three domains: walking distance, walking speed and stair-climbing. Response options for all items comprise a five-point Likert scale ranging from "unable to do" to "no difficulty". Domain scores are generated by multiplying the score for each item by a weighting factor based on the degree of difficulty and then summing all the products together. Scores are then divided by the maximum score of the respective domain to obtain a percentage score, from 0% (i.e. fully impaired) to 100% (i.e. not impaired). WIQ scores are strongly correlated with maximum walking distance [39, 40], objective measures of walking impairment [41], as well as the ankle-brachial index [42].

**Generic HRQoL: SF-12.** The SF-12 is a self-assessment questionnaire with 12 items measuring generic HRQoL [43]. The instrument covers eight health domains: physical functioning, role limitations due to physical health problems, bodily pain, general health, vitality, social functioning, role limitations due to emotional problems and mental health. These domains result in two summary measure scores: Physical Component Summary and the Mental Component Summary. Summary measure scores range from 0 (lowest HRQoL) to 100 (highest HRQoL). The SF-12 is a short version of the SF-36 with good psychometric properties [43]. The SF-12 has been used extensively in both cross-sectional and longitudinal PAD studies to assess (changes in) health status [e.g. 44, 45]. The Mental Component Summary score of the SF-12 was shown to be a valid measure of mental health in the general population [46, 47]. Furthermore, the Physical Component Summary score is associated with PAD severity as measured by the ankle-brachial index [45].

**Depressive symptoms: PHQ-9.** The PHQ-9 is a depression symptom screening instrument with 9 items asking patients how much they were bothered by symptoms over the last two weeks, with scores on a 4-point scale from 'not at all' to 'nearly every day'. The sum score ranges from 0 to 27 and indicates the degree of depression. Scores of $\geq 5$, $\geq 10$, and $\geq 15$ represent mild, moderate, and severe levels of depression, respectively. A cut-off score of $\geq 10$ was found to have a sensitivity and specificity of 0.88 for detecting clinical depression [48].

**Anxiety symptoms: GAD-7.** The GAD-7 is a 7-item screening instrument assessing the core symptoms of generalized anxiety disorder (GAD) over the past two weeks. The answer options are identical to the PHQ-9, with scores on a 4-point scale from 0 ('not at all') to 3 ('nearly every day'). The sum score ranges from 0 to 21 with cut-off scores of 5, 10 and 15 representing mild, moderate and severe levels of anxiety, respectively. A cutoff score of $\geq$ 10 has a sensitivity of 89% and a specificity of 82% for identifying GAD [49]. A study conducted in the German general population confirmed the instrument's good psychometric properties [50].

**Tobacco smoking: Fagerström Test.** The Fagerström Test for Nicotine Dependence [51] is a screening instrument assessing nicotine dependence with respect to cigarette smoking. It consists of six items which are either scored from 0 to 3 (multiple choice items) or 0 to 1 (yes/no items). The sum score ranges from 0 to 10 with higher scores indicating more intense tobacco dependence. The FTND is a revised version of the Fagerström Tolerance Questionnaire (FTQ) [52] with equally good psychometric properties [53]. The Fagerström Test was completed only by those who identified themselves as smokers (n = 668).

**Alcohol consumption: AUDIT-C.** The Alcohol Use Disorders Identification Test-Consumption [54] is a short screening instrument consisting of the original AUDIT's first three items assessing alcohol consumption. Items are scored on a scale ranging from 0 to 4. As a result, the sum score ranges from 0 to 12. Cut-off scores of 3 and 4 are used for women and men to identify patients at increased risk for alcohol-related disorders, whereas a cut-off score of 4 or 5 indicates risky consumption. Research confirmed the validity of the AUDIT-C and found equally good psychometric properties in the AUDIT-C as in the original version [55].

**Other measures.** Along with patient-reported outcome measures (PROMs), sociodemographic and biological variables (age, sex, height, weight, body mass index, education level, household income/economic status, marital status and employment status) and comorbidities (hyperlipidemia, diabetes mellitus, hypertension, lung disease and reduced kidney function) were self-reported during the baseline assessment.

## Statistical analyses

Secondary analyses were performed using cross-sectional baseline data from a parent RCT (i.e. before study interventions were implemented; S1 Fig). Missing data in PROM items (i.e. incomplete information collected from a respondent) were handled via the Expectation-Maximization (EM) imputation algorithm in IBM SPSS Statistics 25, which is an effective and straightforward maximum likelihood technique to manage incomplete data so that there was be no systematic losses of participants who missed single items [56]. EM is recommended to be used in structural equation modeling [57]. An EM estimator is unbiased and efficient when the missing mechanism is missing completely at random or missing at random [58]. Moreover, the EM algorithm is effective when variables had up to 30% missing values. Consequently, participants that had an item nonresponse of > 30% were removed from the analysis dataset (n = 9). Accordingly, data from 1 687 PAD patients were used for this study. In addition, several PROMs were compared against normative data, which were available from previous studies (GAD-7 [50]; PHQ-9 [59, 60]; SF-12 [61]).

Taking consideration of existing literature, a theory-driven path analysis with full information maximum likelihood estimation was conducted to estimate the simultaneous interrelationships between the variables of interest (treated as continuous), while adjusting a priori for empirically identified confounders (i.e. in path models at p < .05: sociodemographic variables, body mass index, comorbidities). Path analysis, which is an extension of multiple regression analysis, is a type of structural equation modeling to clarify (potentially causal) relationships between the variables assessed. The absolute and relative goodness of fit of the models were

assessed based on standard measures of fit; the CMIN/DF statistic (i.e. normed chi-square), the Tucker–Lewis index (TLI), the comparative fit index (CFI), and the root mean square error of approximation (RMSEA). CMIN/DF < 3 indicates an acceptable fit between hypothetical model and sample data [62]. The cut-off for good fit for TLI and CFI is ≥ 0.95 and ≥ 0.90. With values closer to 0 representing good fit, the cut-off for RMSEA, which is an absolute measure of fit, is < .08, indicating excellent fit between model specification and the observed data [63, 64]. Based on model fit indicators, the original model was modified by iteratively including and/or constraining paths and correlations. To arrive at a good model fit, model substructures were assessed based on (standardized) regression weights (i.e. magnitude and p < .05), residual error covariance and modification indices. Several iterations were carried out to arrive at the final revised model. Statistical analyses were performed using IBM SPSS Statistics 25 and SPSS Amos (IBM Corporation, Armonk, New York, United States). Effect sizes (i.e. magnitude of the relationships between path model parameters) were based on Pearson's r correlation coefficients and the standardized beta coefficient (β), with small, medium and large effect sizes indicated by the following r's/ β's, respectively: .10, .30, .50. Effect sizes < .10 are considered negligible regardless of statistical significance in order to avoid possible overinterpretation of small effects.

## Results

### Study sample characteristics

Demographic, socioeconomic and clinical characteristics of the study sample are displayed in Table 1. Most patients were male (67.6%) with a mean age of 66.3 years (SD = 8.6 years; range: 35–81). 64.3% were married, and 81.6% had at least one child. The three most common self-reported comorbidities were hypertension (72.4%), high cholesterol (57.1%) and diabetes mellitus (25.9%). A number of patients had a history of a cardiovascular event; 12.9% of PAD patients suffered from a myocardial infarction (i.e. heart attack), 8.8% had a stroke in the past. Most patients received medication to help control their PAD and cardiovascular comorbidities, such as platelet function inhibitors (81.1%), antihypertensive agents (74.1%) or statins (58.2%). 29.5% underwent revascularization surgery.

### Descriptive statistics

**Mental burden.** The estimated proportion of patients with mild depressive symptoms (PHQ-9 score 5–9) was 30.2% (n = 509). Moderate or severe depressive symptoms, which has been reported to indicate clinical depression, were found in 18.1% (n = 304) of patients (PHQ-9 score ≥ 10). Mild anxiety symptoms (GAD-7 score 5–9) were found in 25.8% (n = 435) of patients, while 9.7% (n = 164) had moderate to severe anxiety symptoms (GAD-7 score ≥ 10) and thus showed signs of clinical anxiety. Compared to the general German population [50, 59, 60], the study population showed substantially higher levels of depression (approx. 84th percentile), and anxiety symptoms (approx. 69th percentile).

**Health risk behavior.** Among the smokers subgroup (n = 668, 39.6% of study participants), low or moderate level of tobacco dependence (Fagerström score 0–4) was identified in 56.3% of patients (n = 376, 22.3% of study participants). The estimated proportion of patients with high or very high tobacco dependence among smokers (Fagerström score 5–10) was 43.7% (n = 292, 17.3% of study participants). Regarding alcohol use, 16.1% (n = 272) were at risk of alcohol-related disorder (AUDIT-C score = 3 in women; = 4 in men), while 29.1% (n = 491) were screened positive for risky consumption (AUDIT-C score ≥ 4 in women; ≥ 5 in men).

**Table 1. Characteristics of the study sample.**

| Variable | n / M | % / SD | range |
|---|---|---|---|
| **Sociodemographic characteristics** | | | |
| **Gender** | | | |
| Female | 525 | 31.1 | |
| Male | 1 141 | 67.6 | |
| Age (in years) | 66.29 | 8.63 | 35–81 |
| **Body Mass Index** | 28.07 | 5.04 | 15–76 |
| **Comorbidities/Diseases** (multiple choices possible) | | | |
| Myocardial infarction | 217 | 12.9 | |
| Stroke | 148 | 8.8 | |
| Metabolism disorder (high cholesterol) | 964 | 57.1 | |
| Angina pectoris | 224 | 13.3 | |
| Lung disease | 270 | 16.0 | |
| Heart Failure | 258 | 15.3 | |
| Hypertension | 1 221 | 72.4 | |
| Diabetes | 437 | 25.9 | |
| Cancer | 155 | 9.2 | |
| **Drugs** (multiple choices possible) | | | |
| Antihypertensive agents | 1 250 | 74.1 | |
| Platelet function inhibitor | 1 368 | 81.1 | |
| Statins | 981 | 58.2 | |
| Other | 918 | 54.4 | |
| **Revascularization** | | | |
| Yes | 498 | 29.5 | |
| No | 931 | 55.2 | |
| **PROMs** | | | |
| **Walking Impairment (WIQ)** | | | |
| Walking distance | 55.75 | 29.20 | 0–100 |
| Walking speed | 48.84 | 24.93 | 0–100 |
| Stair Climbing | 59.66 | 25.88 | 0–100 |
| **Generic HRQoL (SF-12)** | | | |
| Physical | 37.92 | 9.78 | 12–61 |
| Mental | 51.03 | 11.18 | 13–70 |
| **Mental health** | | | |
| **Depressiveness (PHQ-9)** | | | |
| Mild (5–9) | 509 | 30.2 | |
| Moderate(10–14) | 212 | 12.6 | |
| Severe (15–27) | 92 | 5.5 | |
| **Generalized Anxiety (GAD-7)** | | | |
| Mild (5–9) | 435 | 25.8 | |
| Moderate(10–14) | 119 | 7.1 | |
| Severe (15–21) | 45 | 2.7 | |
| **Health Risk behavior** | | | |
| **Alcohol consumption (AUDIT-C)** | | | |
| Risk of alcohol related disorder (= 3 in women; = 4 in men) | 272 | 16.1 | |
| Risky consumption (≥4 in women; ≥5 in men) | 491 | 29.1 | |
| **Tobacco dependence (Fagerström, n = 668)** | | | |
| Low (0–2) | 156 | 23.4 | |

(*Continued*)

**Table 1.** (Continued)

| Variable | n / M | % / SD | range |
|---|---|---|---|
| Moderate (3–4) | 220 | 32.9 | |
| Strong (5–6) | 180 | 26.9 | |
| Very strong dependence (7–10) | 112 | 16.8 | |

Some participants did not provide complete information; the number of responses (n) may therefore deviate from the final dataset (N = 1 687).

**HRQoL.** In comparison to the non-clinical German population [61], physical aspects of HRQoL among PAD patients appeared to be severely impaired (approx. 17th percentile; SF-12 physical composite score: M = 37.92, SD = 9.78), whereas mental aspects of HRQoL were only mildly compromised (approx. 45th percentile; SF-12 mental composite score M = 51.03, SD = 11.18).

## Path analysis

**Original and revised model through respecification.** A theory-driven path analysis (Fig 2) was performed to evaluate the interrelationships between walking impairment (WIQ), HRQoL (SF-12 mental and physical composite scores), health risk behavior (AUDIT-C; Fager-ström-Test) and mental health measures (GAD7; PHQ9). The original model failed to show a sufficiently good fit (CMIN/DF = 10.223; TLI = .942; CFI = .981; RMSEA = .074). As a next step, the model was controlled for potentially confounding effects (i.e. sociodemographic factors, body mass index, comorbidities) and respecified accordingly; goodness-of-fit indices indicated to include age and sex as confounders (represented by grey lines in Fig 2). After controlling for confounding effects, the goodness-of-fit indices showed a better fit to the observed data (CMIN/DF = 7.506; TLI = .939; CFI = .982; RMSEA = .062). Modification indices and residual variances indicated to add one path to the original model, as represented by a red line in Fig 2, which further improved the overall model fit (CMIN/DF = 2.743; TLI = .984; CFI = .996; RMSEA = .032). As a last step, two paths and one error covariance that were not significant were constrained to 0 (represented by thin lines in Fig 2). The results from all model

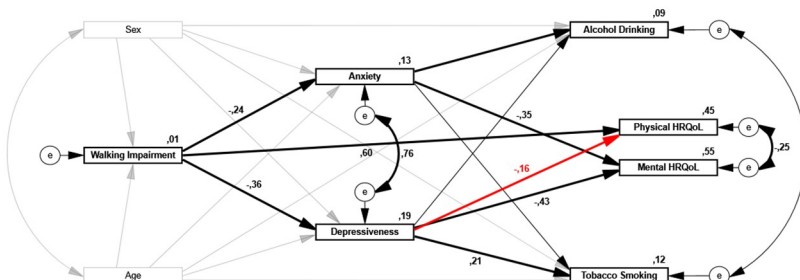

**Fig 2. Final path analysis model testing the influence of walking impairment on mental burden, health risk behavior, physical and mental HRQoL in PAD patients (N = 1 687).** The numbers on the arrows are standardized regression coefficients (β) that indicate the magnitude of effects between variables (coefficients < .10 not shown). Pathways are represented by thick lines, pathways that were constrained to 0 due to model respecification are represented by thin lines. Grey lines represent added covariates to the model (i.e. age, sex) for adjustment of confounding (coefficients not shown; see Table 2). The numbers above the boxes indicate the total proportion of variance explained in the model (R²).

**Table 2. Decomposition of direct effects from the path analysis (Regression and error covariances).**

| Effect | B / Cov | S.E. | β / r | z | p | R² |
|---|---|---|---|---|---|---|
| On Walking Impairment | | | | | | .013 |
| Of Sex | 5.369 | 1.261 | **.104** | 4.257 | *** | |
| Of Age | -.156 | .068 | -.056 | -2.293 | .022 | |
| On Anxiety Symptoms | | | | | | .133 |
| Of Sex | -.792 | .203 | -.090 | -3.909 | *** | |
| Of Age | -.120 | .011 | **-.254** | -11.048 | *** | |
| Of Walking Impairment | -.040 | .004 | **-.237** | -10.395 | *** | |
| On Depressiveness | | | | | | .189 |
| Of Sex | -.960 | .224 | -.096 | -4.282 | *** | |
| Of Age | -.120 | .012 | **-.221** | -9.964 | *** | |
| Of Walking Impairment | -.069 | .004 | **-.356** | -16.104 | *** | |
| On Physical HRQoL | | | | | | .446 |
| Of Depressiveness | -.328 | .040 | **-.157** | -8.116 | *** | |
| Of Walking Impairment | .243 | .008 | **.596** | 31.732 | *** | |
| On Mental HRQoL | | | | | | .549 |
| Of Anxiety Symptoms | -.955 | .071 | **-.350** | -13.434 | *** | |
| Of Depressiveness | -1.031 | .063 | **-.432** | -16.388 | *** | |
| On Alcohol Drinking | | | | | | .092 |
| Of Sex | 1.512 | .123 | **.291** | 12.315 | *** | |
| Of Age | -.022 | .007 | -.079 | -3.250 | .001 | |
| Of Anxiety Symptoms | -.035 | .014 | -.059 | -2.420 | .016 | |
| On Tobacco Smoking | | | | | | .119 |
| Of Sex | .381 | .180 | .078 | 2.114 | .035 | |
| Of Age | -.063 | .010 | **-.239** | -6.372 | *** | |
| Of Depressiveness | .101 | .018 | **.208** | 5.549 | *** | |
| Covariance Between Sex | | | | | | |
| And Age | .424 | .099 | **.106** | 4.290 | *** | |
| Error Covariance Between Depressiveness | | | | | | |
| And Anxiety Symptoms | 12.208 | .490 | **.764** | 24.906 | *** | |
| Error Covariance Between Mental HRQoL | | | | | | |
| And Physical HRQoL | -13.726 | 1.366 | **-.252** | -10.047 | *** | |

Table columns: unstandardized beta (B) or covariance (Cov) coefficient, the standard error for the unstandardized beta or covariance (S.E.), the standardized beta (β) or Pearson correlation coefficient (r), the critical ratio (z), and the probability value (p; *** < .001). β / r >.10 are highlighted in bold.

iterations are presented in S2 File. Following these iterations, the goodness-of-fit indices showed an excellent fit between the observed data and the revised model (CMIN/DF = 2.345; TLI = .987; CFI = .996; RMSEA = .028). The final revised model, with accompanying path coefficients (i.e. standardized regression weights) and squared multiple correlations, is presented in Fig 2. The results of the decomposition of effects based on the path analysis model are shown in Table 2.

**Walking impairment, mental health burden and health risk behavior.** Limitations due to IC symptoms were associated with an overall increase in mental burden; in the model, walking impairment had a moderate direct effect on depressive symptoms (β = -.36) and, to a lesser extent, on anxiety symptoms (β = -.24). The model explained 19% and 13% of the variance in depressive and anxiety symptoms, with age and sex accounting for a large portion of the total explained variance (Table 2).

Depressive symptoms, in turn, had a small to moderate direct effect on the amount of tobacco smoking (β = .21), while anxiety symptoms had a negligible direct effect on alcohol drinking (β = -.06). The model explained 12% and 9% of the variance in tobacco and alcohol use, but again with age and sex accounting for the large portion of the explained variance. The indirect effects of walking impairment on tobacco and alcohol use mediated through mental health burden (i.e. depressive and anxiety symptoms) are considered negligible (total indirect β's < .10).

**Walking impairment on health-related quality of life.** Walking impairment had a large direct effect on lower physical HRQoL (β = .60). In addition, walking impairment was also substantially predictive of lower mental HRQoL, although fully mediated by an increase in depressive and anxiety symptoms (total indirect β = .24). Overall, the model explained 45% and 55% of the variance in physical and mental HRQoL.

## Discussion

### Mental health

The descriptive results show that patients with PAD are at much higher risk of presenting depressive symptoms compared with the general population [59, 60]. The reported prevalence is in good agreement with previous studies [17]; for instance, depression or depressive symptoms have been observed in 16% [14], 19.6% [13], 21.7% [15], 24% [65], 30% [12], up to 36.1% shortly before revascularization [66], compared to 48.3% showing at least mild depressive symptoms in this study, and of these, 18.1% showings signs of clinical depression (PHQ-9 score > = 10). Finally, results from the path model confirm that young female PAD patients are at modestly higher risk to develop depressive symptoms [17, 29, 67, 68].

Moreover, the path analysis results support the hypothesis that poor walking ability contributes to depressive symptoms [14–17, 65, 69]. These findings support previous studies that show an association between PAD outcomes and depressive symptoms. PAD patients suffering from depressive symptoms are more likely to have other clinical symptoms such as chest discomfort, shortness of breath, and heart palpitations [15]. In addition, comorbid depressive symptoms are associated with an increased risk of PAD events [19], secondary cardiovascular events [22], major amputations [20, 70] and mortality [13]. Finally, PAD patients with depressive symptoms are less willing to exercise [69] and more likely to have recurrent symptoms after revascularization [66], which may further aggravate PAD symptoms. Although the current indicate a single direction of effect, the relationship between depression and PAD outcomes has been hypothesized to be bi-directional in a mutually reinforcing cycle, meaning that PAD outcomes are thought to increase depression and vice versa [71].

The body of literature examining other mental health issues in PAD patients is limited, as much of the literature tends to focus on the relationship between depression and PAD. That being said, the descriptive results show that PAD patients are also at much higher risk of having comorbid anxiety symptoms compared with the general population [50]. Anxiety symptoms were observed in 35.6%, which is comparable to 30% [12] and 24.4% [11] in other studies. The path analysis results support the hypothesis that poor walking ability contributes to anxiety symptoms. As for depression, it is reasonable to assume that anxiety disorders and PAD outcomes are also in a bidirectional relationship, as anxiety has been previously identified as a risk factor for the development of IC [18] and is associated with severe leg symptoms (i.e. pain at rest) [12]. Furthermore, it was shown that anxiety enhances the detrimental effect of depressive symptoms on health status after revascularization [72]. Future studies should further explore this association and whether tailored mental health interventions would improve PAD outcomes.

## Health risk behavior

The descriptive results also demonstrate that PAD patients, especially men of younger age, are at great risk of engaging in health risk behavior such as tobacco use and excessive alcohol consumption. In the present study, 45.2% of patients were identified to be at risk of an alcohol-related disorder or exhibit risky alcohol consumption, compared to 39% found in a previous study [73]. Additionally, 39.6% of PAD patients were identified as smokers. Other research investigating home-based exercise programs in PAD patients reported smoking frequencies ranging from as little as 21.4% [44] up to 86.1% [74], suggesting a considerable heterogeneity between studies. Such health risk behavior can cause profound damage to vascular health. In particular, tobacco use is the single most important cause and leading risk factor of PAD [3] with a threshold- [75], dose-response-relationship [76], fostering progression of functional impairment [77, 78] and thereby resulting in a reduced HRQoL [78]. In terms of alcohol consumption, a U- or J-shape dose-response relationship between alcohol use and PAD has been suggested in various studies, meaning that low-to-moderate alcohol consumption, (particularly of red wine [79]) may result in a reduction of cardiovascular events and mortality [32–34, 80], while heavy/risky drinking is severely detrimental to PAD [31–36].

Mounting evidence suggests that risky health behaviors are part of the underlying mechanistic link between mental health status and PAD outcomes [10]. The path analysis results support the hypothesis that mental distress results in an increased desire to smoke among PAD patients [19], which in turn may contribute to the pathogenesis and deterioration of PAD. Regardless of PAD status, smoking is generally suggested as a self-medication strategy to cope with mental distress [81, 82]. However, there is evidence of a bidirectional relationship between depression and smoking, in which depressive symptoms lead to self-medication by smoking, which in turn causes changes in the dopaminergic system leading to depressed mood [82]—a vicious that eventually leads to a further deterioration of PAD.

One unanticipated result of the study was that mental burden was not associated with an increase in alcohol use. In the general population, the current literature suggests a causal linkage of alcohol use increasing the risk for depression, while alcohol being used to self-medicate symptoms of depression to reduce emotional distress [83]. These findings cannot be confirmed for PAD patients, suggesting that alcohol use plays no major role in the mechanistic relationship between mental distress and negative PAD outcomes. However, it is important to note that patients with a clinically diagnosed affective and/or substance use disorder were excluded from the study, which may have affected the current results. It is possible that severely depressed patients and those suffering severely from anxiety are more likely to show stronger signs of alcohol use disorder [84], which however remains to be investigated in further studies.

## HRQoL

Compared to the general population, PAD patients were also found to have an impaired HRQoL, which is largely consistent with previous studies that have demonstrated poor HRQoL in PAD patients [45, 85, 86]. Similar to previous studies [23, 24, 45, 85], lower HRQoL was more evident for the physical aspects of HRQoL.

Moreover, in accordance with previous findings [23–26], the current path analysis support the hypothesis that poor HRQoL is largely related to increasing walking impairment. For mental HRQoL, this effect was largely mediated by the PAD patient's magnitude of depressive and anxiety symptoms, which highlights that the reduction of HRQoL is to large extent due to IC's detrimental effect on mental health [46]. The detrimental effect of mental health on HRQoL measures has been well demonstrated in previous studies [28, 29]. Similarly, in patients with

chronic heart failure, depression predicted physical and mental aspects of HRQoL [87]. Notably, these effects were independent of other physically debilitating comorbidities that also have an impact on patients' HRQoL (e.g. lung diseases).

## Implications for clinical practice

The path analysis results provide a reasonable explanation of the complex interaction between functional walking limitations, mental burden, health risk behavior and quality of life in PAD patients, holding important clinical and public health implications. First and foremost, improving IC symptoms is a vital treatment approach for PAD patients, since walking impairment was found to be a crucial determinant of mental burden and HRQoL. Because the improvement of HRQoL is a key therapeutic goal in the treatment of PAD patients, staying physically active with appropriate exercise programs and other treatment modalities to improve walking impairment (e.g. revascularization) remains an integral key part of PAD treatment. The secondary benefits of walking improvement on HRQoL through exercise therapy is well established [88–90]. Importantly, to monitor changes in walking impairment as a result of treatment, the treating clinicians should use patient-reported measures of functional disability [25, 26], as these are the primary predictor of HRQoL [25, 26]. In contrast, HRQoL in PAD patients is only marginally associated with clinical markers as well as objective measurements of walking impairment [91–93], or HRQoL assessments of physicians [94], suggesting that the experience of HRQoL is not fully reflected by these surrogate measures. The importance of HRQoL, reflecting the physical, social and mental well-being of PAD patients, is being increasingly recognized by physicians and stakeholders (patients, relatives, etc.) as a significant indicator of treatment success [95, 96], which is nowadays also acknowledged in several international guidelines [5, 97, 98].

Second, the presence of mental health issues has been related to adverse health outcomes and influences the prognosis and treatment response of PAD, which points to the importance of integrating psychological aspects into therapy conversations. To break the self-perpetuating circle between mental health burden and PAD progression (i.e., amputation, cardiovascular events) [13–17, 19–22], vascular care providers should pay close attention to mental health challenges of PAD patients by integrating mental health care providers in an interdisciplinary/ collaborative care model [17], which likely would have secondary benefits in reducing PAD risk and improving HRQoL. In patients with corona artery disease, previous studies have demonstrated the positive impact of depression and anxiety care and stress management on cardiovascular and psychological outcomes [99–101]. Likewise, the treatment of depressive symptoms improve physical functioning in older adults [102], which should therefore be considered a viable therapeutic approach in the management of PAD. Remarkably, there is already a guideline on addressing depression for patients with coronary artery disease [103], recommending to use the PHQ-9 to screen for depression, but not yet for PAD, which should be urgently addressed in the near future as psychosocial stressors play a critical role in PAD development and progression. For instance, psychological distress [104], including work-related stress [105], was found to elevate the risk for PAD and show a poor PAD recovery pathway [29].

Third, reducing health risk behavior should always be a key target of PAD management for effectively reducing adverse PAD outcomes, as tobacco and excessive alcohol use are known to be potent factors in the development and progression of PAD. Changes in health risk behavior should be also addressed through the delivery of PAD lifestyle interventions ([106], e.g. smoking cessation programs), which are usually guided by conceptual frameworks for risk behavior modification (for an overview, see [107]). The use of lifestyle interventions is currently low,

although they have substantial secondary cardiovascular benefits and may prevent further worsening of PAD [106]. Finally, it has been speculated that depressed patients use smoking as a form of self-medication to relieve symptoms; accordingly, modifying risk health behaviors could be achieved by improving mental health. In fact, there is good evidence that psychological interventions are effective in reducing smoking by people with mental health problems [108], which may in turn have a beneficial effect on the patient's PAD status and HRQoL.

## Limitations

Several potential limitations must be mentioned when interpreting the findings of this study. Since all measures were collected cross-sectional, no certain conclusions can be drawn about temporal and causal relationships. Therefore, the directional arrows in the path analysis should be interpreted with great caution, as path analysis cannot prove causality out of a cross-sectional study (which can be proven only through the correct research design), but rather intended to test whether the data are consistent with the postulated causal model based on theoretical considerations. With that said, the current models can help to generate causal hypotheses for future studies, even more so given the "real-world" setting of this study with a study sample that is highly representative of PAD patients with IC.

Furthermore, despite the largely theory-driven approach, it is important to treat some of the results with caution, as some of them show trivial effect sizes although being statistically significant. For path analysis, an adequate sample size should normally be ten times the amount of the parameters as a rule of thumb, whereas in this study the sample is almost 200 times higher, which increases the power of detecting very small effects resulting in statistical significance. Although larger studies such as this one are unquestionably valuable and generally viewed as a favorable development, with all parameter values estimated with higher accuracy, it is important to consider how clinically and practically 'relevant' these effects really are in relation to prior literature. Therefore, to avoid an overinterpretation of effects ('large sample size fallacy'), it is important to interpret all effects in the model appropriately according to their size, and not treat statistical significance synonymously with practical significance [109]. At the same time, the large sample size can be regarded as a strength of the study, as the current findings comprise robust evidence and are most likely not caused by a statistical artifact.

## Conclusions

In conclusion, the present results demonstrate that PAD patients often experience substantial impairment in terms of functional health status, mental burden and HRQoL. In order to better understand the complex relationship between these factors, the study sought to integrate clinical indicators and psychosocial aspects of PAD into a single theoretical model, thereby supporting a more comprehensive, multimodal therapeutic approach to PAD. The findings clearly indicate the importance of not only including somatic health status in the treatment of PAD, but also accounting for psychosocial aspects in PAD. To expand the explanatory ability of the complex relationships in the field of PAD, future research should put greater efforts in a theory-based approach using more sophisticated multivariate data analysis techniques (e.g. structural equation models) that defines the entire set of relationships.

## Supporting information

**S1 Fig. Flow chart of the randomized-controlled trial.**
(TIF)

**S1 File. All relevant study data (SPSS file).**
(SAV)

**S2 File. Model iterations.**
(PDF)

## Acknowledgments

We would like to thank Finja Mäueler, who assisted us with great enthusiasm and commitment in the preparation of the manuscript.

## Author Contributions

**Formal analysis:** Farhad Rezvani, Mara Pelt.

**Funding acquisition:** Martin Härter, Jörg Dirmaier.

**Methodology:** Farhad Rezvani.

**Project administration:** Jörg Dirmaier.

**Supervision:** Martin Härter, Jörg Dirmaier.

**Writing – original draft:** Farhad Rezvani, Mara Pelt.

**Writing – review & editing:** Farhad Rezvani, Martin Härter, Jörg Dirmaier.

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
