## [Decision Letter · Decision Letter 0]

8 Jun 2022

PONE-D-21-26241Effects of walking impairment on mental health burden, health risk behavior and quality of life in patients with intermittent claudication: a cross-sectional path analysis.PLOS ONE

Dear Dr. Rezvani,

Thank you for submitting your manuscript to PLOS ONE. After careful consideration, we feel that it has merit but does not fully meet PLOS ONE’s publication criteria as it currently stands. Therefore, we invite you to submit a revised version of the manuscript that addresses the points raised during the review process.

Although one reviewer made the decision of accept, other reviewer had several concerns about your manuscript. Please revise your manuscript according reviewer's comments.

We look forward to receiving your revised manuscript.

Kind regards,

Kenji Hashimoto, PhD

Section Editor

PLOS ONE

Journal Requirements:

Reviewers' comments:

Reviewer's Responses to Questions

**Comments to the Author**

1. Is the manuscript technically sound, and do the data support the conclusions?

Reviewer #1: Yes

Reviewer #2: Partly

2. Has the statistical analysis been performed appropriately and rigorously? 

Reviewer #1: Yes

Reviewer #2: I Don't Know

3. Have the authors made all data underlying the findings in their manuscript fully available?

Reviewer #1: Yes

Reviewer #2: Yes

4. Is the manuscript presented in an intelligible fashion and written in standard English?

Reviewer #1: Yes

Reviewer #2: Yes

5. Review Comments to the Author

Reviewer #1: This study investigated the effects of walking impairment on mental health burden, health risk behavior and quality

of life in patients with intermittent claudication.

This is a valuable paper for researchers and clinicians.

Although this type of decision outcome is rare, I recommend acceptance of the manuscript without any changes.

I thank the editor and the authors for giving me the opportunity for reading the manuscript.

Reviewer #2: This study aimed to examine the association between walking impairment, health-related quality of life, anxiety/depression, and health behaviors (nicotine/alcohol use).

A theory-driven path model was constructed, with inverse associations reported between walking impairment and mood symptoms, and subsequent tobacco use. These were all small effects. Walking impairment also was associated with lower mental health status, partially mediated through depression/anxiety experiences. The authors are underscoring the need for more comprehensive treatment strategies for PAD.

This is a very relevant message that the authors bring — I have the following feedback:

Abstract

please clarify setting and timeframe of the study, as well as numbers included into the study, and research design

Introduction

Nicely summarized

1) since this is a theory driven pathway analysis, it would be good to list the hypotheses informing your model.

Methods

- since this is a secondary data analysis of a trial, it would be helpful to have more details about the design of the trial, the conditions, and the different assessments and the timing of those assessments. Is there a visual representation of cohort derivation and assessments that can be provided?

- Why was the SF-12 chosen as the main health status instrument? Since this is a very generic instrument, it may not be as sensitive to the functioning that is specific and relevant to PAD

- I don’t understand the statement that says (line 135) “with regard to the mental scale, it was shown that it validly reflects depressiveness and anxiety disorders”

- It is unclear at which time points the assessments were done and how the intervention vs. control cohorts were different from each other and how the intervention could have impacted the outcomes/assessments, nor what the duration of follow up was.

- Missingness was handled by either elimination or imputation. Can more information be provided about the imputation algorithm and why not multiple imputation was considered? How were people with vs. without missing information different from each other? How were deaths handled?

- How were the iterations of the path model before and evaluated?

- Were the variables used as continuous variables?

- What interpretation guidelines were adopted to evaluate whether the effect sizes of the beta coefficients?

Results

- there seem to be some ad hoc adjustment to the theory based model with confounders added. It would be good to have the choice of confounders also be a priori considered about the theory driven model. It is unclear to me what criteria were used for the different iterations, other than the RMSEA values.

- Table 2 presents the regression results, but again, not clear as to what the time frames were and how many models were created. It is hard to follow along.

- After table 2, the results section becomes very hard to follow. All effects are given equal prominence, whereas some beta coefficients are clearly very low and may not be as relevant to emphasize.

- It is also unclear which aspects of the walking impairment was most associated with outcomes

Discussion

- The discussion seems to be a mix about the prevalence of the symptoms noted as well as a commentary on some of the health behaviors. I see less of a reflection on some of the directionality and targets for intervention, informed by the path modeling, other than that we should address smoking and mental health burden.

- I think a more focused discussion (after more structuring of the results) may be helpful to understand what novel insights have been derived, other than establishing the associations. It is also hard to judge the implications of the analyses without have a clear timeframe of the assessments and whether or not the assessments were repeated at intervals so as to re-construct patients’ pathways.

6. PLOS authors have the option to publish the peer review history of their article (what does this mean?). If published, this will include your full peer review and any attached files.

Reviewer #1: No

Reviewer #2: No

---

## [Author Response · Author response to Decision Letter 0]

5 Aug 2022

--We would like to thank the editor and the reviewers for their useful feedback. We are happy that they overall find the paper informative. We have addressed all comments and revised the manuscript accordingly.

Please find our responses to each comment below. Revisions in the manuscript were tracked in response to each point.

Journal Requirements:

--Response: The following information has been added to the Data Availability statement: “All relevant data are included in the Supporting Information file S2 (SAV).”

Review Comments to the Author

Reviewer #1: 

This study investigated the effects of walking impairment on mental health burden, health risk behavior and qualityof life in patients with intermittent claudication. This is a valuable paper for researchers and clinicians. Although this type of decision outcome is rare, I recommend acceptance of the manuscript without any changes. I thank the editor and the authors for giving me the opportunity for reading the manuscript.

--Response: We are very pleased with the positive feedback from Reviewer #1. Thank you!

Reviewer #2:

This study aimed to examine the association between walking impairment, health-related quality of life, anxiety/depression, and health behaviors (nicotine/alcohol use). A theory-driven path model was constructed, with inverse associations reported between walking impairment and mood symptoms, and subsequent tobacco use. These were all small effects. Walking impairment also was associated with lower mental health status, partially mediated through depression/anxiety experiences. The authors are underscoring the need for more comprehensive treatment strategies for PAD.

This is a very relevant message that the authors bring — I have the following feedback:

Abstract

Please clarify setting and timeframe of the study, as well as numbers included into the study, and research design

Response: Good point, thank you. The timeframe of data collection (4/2017-12/2017), the number of patients included in the analysis (N=1696) and the original research design (randomized controlled trial) has been added to the revised abstract.

Introduction

Nicely summarized

--Response: Thank you!

Since this is a theory driven pathway analysis, it would be good to list the hypotheses informing your model.

--Response: We agree that listing the hypotheses would be useful to the reader in assessing the theory-based model. A brief description of the postulated model, along with the corresponding hypotheses, has been added to the revised introduction.

Methods

Since this is a secondary data analysis of a trial, it would be helpful to have more details about the design of the trial, the conditions, and the different assessments and the timing of those assessments. Is there a visual representation of cohort derivation and assessments that can be provided?

--Response: The previous version did not specify that only cross-sectional baseline data from the RCT were used. This information has now been added to the abstract and method section (Design and Statistical analyses). We have also included a flowchart as Supporting Information to illustrate which RCT data we used for this study (baseline assessment). Because only baseline data were used (i.e. pre-intervention), a description of the study arms and follow-up time points is not considered necessary. 

Following the reviewer's suggestion, we have provided more background information about the recruitment process, allowing the reader to understand how the study population was derived. In addition, as noted in the manuscript, further details about the methods of the RCT (i.e. design, conditions, assessments, cohort derivation) are provided in the study protocol (Rezvani, Heider et al., 2020 in BMJ Open).

Why was the SF-12 chosen as the main health status instrument? Since this is a very generic instrument, it may not be as sensitive to the functioning that is specific and relevant to PAD

--Response: Despite being a generic health status instrument, the SF-12 and the longer versions SF-20/36 have been used extensively in both cross-sectional and longitudinal PAD studies. Furthermore, the SF-12 has been shown to validly capture health status (Wu, Coresh, Selvin, Tanaka, Heiss, Hirsch et al., 2017) and changes in health status in PAD patients (e.g. McDermott Guralnik Criqui Ferrucci Zhao et al., 2014). For clarity, this information has been added to the revised method section.

Although the VascuQoL-25 with five subscales was also collected, the SF-12 was used for the present research question to keep the model simple and allow straightforward conclusions to be drawn regarding the effect walking impairment on (mental and physical) quality of life.

I don’t understand the statement that says (line 135) “with regard to the mental scale, it was shown that it validly reflects depressiveness and anxiety disorders”

--Response: We agree that the original wording was somewhat misleading. The sentence was revised accordingly.

It is unclear at which time points the assessments were done and how the intervention vs. control cohorts were different from each other and how the intervention could have impacted the outcomes/assessments, nor what the duration of follow up was.

--Response: The study was cross-sectional in nature with no use of follow-up assessments from the RCT (baseline data, i.e. before study interventions were implemented. For clarity this information has been added to the abstract and the revised method section. In addition, we have included a flowchart as Supporting Information to illustrate which RCT data we used for this study.

Missingness was handled by either elimination or imputation. Can more information be provided about the imputation algorithm and why not multiple imputation was considered? How were people with vs. without missing information different from each other? How were deaths handled?

--Response: Expectation maximization was employed only for the purpose of imputing missing items. EM, which is an algorithm for computing maximum likelihood estimates, is an effective and straightforward technique to manage incomplete data at the item level (i.e. incomplete information collected from a respondent); see Dempster, Laird, & Rubin, 1977. As with Multiple Imputation, an EM estimator is unbiased and efficient when the missing mechanism is ignorable (i.e. missing completely at random or missing at random), see Dong & Peng, 2013 for further clarification. Moreover, it should be noted that the percentage of missing values in each column was approximately 3%, which means that the risk of introducing bias due to EM is considered negligible.

EM is commonly used in exploratory path models, e.g. Wurm, Tomasik, & Tesch-Römer, 2010; Schotanus-Dijkstra, Pieterse, Drossaert, Walburg, & Bohlmeijer, 2019) and is recommended when a single dataset is sought for exploratory path analysis (Hair, 2009). Following the reviewer's suggestion, we have added further details about EM to the statistical analysis section.

Deaths occurring during the RCT period were not relevant for this study as data were only analyzed cross-sectionally (i.e. baseline data only).

How were the iterations of the path model before and evaluated?

--Response: Changes to the model were based on the inspection of modification indices, and regression weights (i.e. magnitude and p < .05). Between each iteration, several goodness-of-fit parameters were examined until achieving satisfactory model fit (RMSEA, CMIN/DF statistic, Tucker–Lewis index, comparative fit index; see statistical analysis section).

In response to your comment, we have included all model iterations as supplementary material to make the sequence of model decisions more transparent.

Were the variables used as continuous variables?

--Response: All variables were treated as continuous variables. For clarity this information has been added to the method section.

What interpretation guidelines were adopted to evaluate whether the effect sizes of the beta coefficients?

--Response: The magnitude of the relationships between path model parameters were based on Pearson’s r correlation coefficients (equivalent to standardized beta coefficients), with small, medium and large effect sizes indicated respectively by the following r’s: .10, .30, .50. This information has been added to the method section. In addition, effect sizes <.10 are considered negligible and are not displayed in the final reversed model (Fig2) to avoid overinterpretation of small effects.

Results

There seem to be some ad hoc adjustment to the theory based model with confounders added. It would be good to have the choice of confounders also be a priori considered about the theory driven model. 

--Response: Thank you for your valuable comment. Following your recommendation, we decided to insert the covariates a priori into the model to ensure that the relationship between two variables is not confounded/exaggerated. This resulted in an slightly improved model fit overall, with a minor change in the model. Because of the very small effect, this change has very little explanatory power, meaning that the modification of the model has no influence on the overall interpretation of the results. 

It is unclear to me what criteria were used for the different iterations, other than the RMSEA values.

--Response: Changes to the model were based on the inspection of modification indices, and regression weights (i.e. magnitude and p < .05). Between each iteration, several goodness-of-fit parameters were examined until achieving satisfactory model fit (RMSEA, CMIN/DF statistic, Tucker–Lewis index, comparative fit index; see statistical analysis section). The iteration procedure is described under the section Path analysis -- Original and revised model through respecification.

In addition, we have included all model iterations as supplementary material to make the sequence of model decisions more transparent.

Table 2 presents the regression results, but again, not clear as to what the time frames were and how many models were created. It is hard to follow along.

--Response: The cross-sectional path model analysis was based on baseline data from a RCT (i.e. only one study point). For clarity this information has been added to the abstract and the revised method section. In addition, we have included a flowchart as Supporting Information to illustrate which RCT data was used for this study.

In the updated version of the manuscript, the original model included the hypothesized paths, with age and sex included as confounding variables, visualized in Figure 1. The second model included an additional path from depressiveness to physical health-related quality of life. Finally, two non-significant paths as well as one error covariance were constrained to zero. Please note that we have also included all model iterations as supplementary material to make the sequence of model decisions more transparent.

After table 2, the results section becomes very hard to follow. All effects are given equal prominence, whereas some beta coefficients are clearly very low and may not be as relevant to emphasize.

--Response: Thank you for your valuable comment. We have revised the results section with more emphasis on the relevant effects based on the formulated hypotheses and the proposed model, making it much simpler for the reader to understand what has been found. Additionally, effect sizes <.10 are considered negligible regardless of statistical significance in order to avoid possible overinterpretation of small effects. Finally, the effects of age and sex, which had been identified as confounders and included in the model, were removed from the results section.

It is also unclear which aspects of the walking impairment was most associated with outcomes

--Response: Although this is an interesting research topic in its own right, we believe it is beyond the scope of this paper. In particular, there is a lack of literature that would have allowed to postulate hypotheses about the different aspects of walking impairment, which would have inevitably led to a more exploratory, data-driven rather than hypothesis-informed construction of the a priori model. 

That is, including the three subscales of the WIQ (walking distance, walking speed, stair-climbing) in the model instead of using the total score would have made the theory-based development, analysis and interpretation of the model much more complex. In our opinion, the major advantage of the current model lies in its simplicity with straightforward clinical implications for the treatment of PAD. 

Discussion

The discussion seems to be a mix about the prevalence of the symptoms noted as well as a commentary on some of the health behaviors. I see less of a reflection on some of the directionality and targets for intervention, informed by the path modeling, other than that we should address smoking and mental health burden.

--Response: Thank you for your valuable comment. Following your comment, we have focused more closely on the results of the path analysis by discussing the potential directions of effects and its direct clinical implications by proposing targets for intervention. That being said, we are very cautious about making strong assumptions regarding the directionality of the effects, as this is a cross-sectional path analysis that precludes conclusions about causal/temporal relationships. 

Conclusions that were made based on very small effects were removed from the discussion. Furthermore, for a better understanding, we have divided the discussion into four subheadings: 1) Discussion; 2) Implications for clinical practice; 3) Limitations; and 4) Conclusion. 

I think a more focused discussion (after more structuring of the results) may be helpful to understand what novel insights have been derived, other than establishing the associations. It is also hard to judge the implications of the analyses without have a clear timeframe of the assessments and whether or not the assessments were repeated at intervals so as to re-construct patients’ pathways.

--Response: Following your comment, we have revised the results and discussion section accordingly. After more structuring of the results, we have focused more closely on the results of the path analysis and its direct clinical implications.

We apologize for the confusion regarding the timeframe of the assessments; as described in the methods section, this is a cross-sectional study based on baseline data from RCT (i.e. pre-intervention).

We thank Reviewer #2 for his/her careful reading and insightful questions, which we feel have improved the paper.

---

## [Decision Letter · Decision Letter 1]

16 Aug 2022

Effects of walking impairment on mental health burden, health risk behavior and quality of life in patients with intermittent claudication: a cross-sectional path analysis.

PONE-D-21-26241R1

Dear Dr. Rezvani,

We’re pleased to inform you that your manuscript has been judged scientifically suitable for publication and will be formally accepted for publication once it meets all outstanding technical requirements.

Kind regards,

Kenji Hashimoto, PhD

Section Editor

PLOS ONE

Additional Editor Comments (optional):

Reviewers' comments:

Reviewer's Responses to Questions

**Comments to the Author**

1. If the authors have adequately addressed your comments raised in a previous round of review and you feel that this manuscript is now acceptable for publication, you may indicate that here to bypass the “Comments to the Author” section, enter your conflict of interest statement in the “Confidential to Editor” section, and submit your "Accept" recommendation.

Reviewer #1: All comments have been addressed

2. Is the manuscript technically sound, and do the data support the conclusions?

Reviewer #1: Yes

3. Has the statistical analysis been performed appropriately and rigorously? 

Reviewer #1: Yes

4. Have the authors made all data underlying the findings in their manuscript fully available?

Reviewer #1: Yes

5. Is the manuscript presented in an intelligible fashion and written in standard English?

Reviewer #1: Yes

6. Review Comments to the Author

Reviewer #1: (No Response)

7. PLOS authors have the option to publish the peer review history of their article (what does this mean?). If published, this will include your full peer review and any attached files.

Reviewer #1: No

---

## [Editor Report · Acceptance letter]

24 Aug 2022

PONE-D-21-26241R1 

Effects of walking impairment on mental health burden, health risk behavior and quality of life in patients with intermittent claudication: a cross-sectional path analysis. 

Dear Dr. Rezvani:

I'm pleased to inform you that your manuscript has been deemed suitable for publication in PLOS ONE. Congratulations! Your manuscript is now with our production department. 

Kind regards, 

on behalf of

Prof. Kenji Hashimoto 

Section Editor

PLOS ONE